# Molecular Design and Role of the Dynamic Hydrogen Bonds and Hydrophobic Interactions in Temperature-Switchable Polymers: From Understanding to Applications

**DOI:** 10.3390/polym17111580

**Published:** 2025-06-05

**Authors:** Yurij Stetsyshyn, Halyna Ohar, Andrzej Budkowski, Giuseppe Lazzara

**Affiliations:** 1Department of Organic Chemistry, Lviv Polytechnic National University, 3/4 St. George’s Sq., 79013 Lviv, Ukraine; 2Department of Civil Safety, Lviv Polytechnic National University, 1 St. George’s Sq., 79013 Lviv, Ukraine; halyna.o.ohar@lpnu.ua; 3Department of Molecular and Interfacial Biophysics, Smoluchowski Institute of Physics, Jagiellonian University, Łojasiewicza 11, 30-348 Kraków, Poland; andrzej.budkowski@uj.edu.pl; 4Dipartimento di Fisica e Chimica, Università degli Studi di Palermo, Viale delle Scienze pad 17, 90128 Palermo, Italy

**Keywords:** stimuli-responsive polymers, LCST, UCST, hydrogen bonds

## Abstract

Temperature-induced transitions in polymer systems, often governed by a phenomenon called critical solution temperatures (CSTs), lie on the basis of various advanced technologies such as tissues detachment, smart windows, enhanced DNA biosensors, etc. Despite this application-oriented progress, the molecular mechanisms of the temperature-induced transition based on CSTs remain often underexplored or weakly explained. In this review, we focus on the different molecular mechanisms driving CST-based transitions, systematizing information on homofunctional polymer systems. Understanding these mechanisms is crucial for manipulating temperature-sensitive properties, which offers significant potential for future innovations in smart materials.

## 1. Introduction

Temperature-responsive polymers are a broad group of polymers with various mechanisms of temperature-induced responses [1]. These polymers have the unique ability to reversibly change their physicochemical properties within relatively narrow temperature ranges [2,3,4]. Currently, the classification of temperature-responsive polymers includes transitions from a glassy or crystalline state to a rubbery state [5,6,7,8], from a nematic state to an isotropic state [9,10,11,12], reversible temperature-induced covalent bonding between complementary monomeric subunits [13,14], and the coil–globule transition in aqueous solutions, which occurs at the lower critical solution temperature (LCST), or conversely, from globule to coil at the upper critical solution temperature (UCST) [1,15,16]. Currently, the most widely studied and applied systems are those based on critical solution temperatures. In such systems, a phase transition across the critical solution temperature is accompanied by a change in the polymer conformation [2]. The phenomenon of lower critical solution temperature (LCST) in polymers was first reported by Heskins and Guillet using poly(N-isopropylacrylamide) (PNIPAM) in aqueous solutions [17]. This work was expanded by Halperin et al., who provided a comprehensive review of phase diagrams for various PNIPAM systems [18]. In turn, Silberberg et al. performed temperature-dependent light scattering experiments in dilute aqueous solutions of poly(methacrylamide) (PMAAm), identifying a theta point in water at 6 °C. Their findings suggested the presence of both intramolecular and intermolecular hydrogen bonds among PMAAm fragments. However, despite these indications of potential phase behavior, UCST-type phase separation was not observed [19]. A significant advancement in the study of temperature-responsive systems was the demonstration of both LCST and UCST behavior in a single polymer system in an aqueous solution. Partially butyralized poly(vinyl alcohol), with a substitution degree of approximately 7.5 mol%, showed LCST at 25 °C and UCST at 135 °C [20].

Polymers with critical solution temperatures (CSTs or TCs) form the most studied group of temperature-responsive polymers and have led to the development of breakthrough technologies. For example, T. Okano and colleagues developed the commercially available Nunc™ Multidishes with UpCell^TM^ Surface [21,22]. These dishes feature grafted polymer brushes with LCST properties, which are adhesive to cells at 37 °C but become antifouling when the temperature drops below the LCST. Hence, cells adhere and grow under optimal conditions to form a cell sheet released upon temperature stimuli. Traditionally, cells and tissues are detached from scaffolds using trypsin, which can cause partial tissue damage. However, Okano’s approach eliminates the need for trypsin, preserving the integrity of the tissue (see Figure 1a). Another application of CST-responsive polymers is in smart windows (Figure 1b). Studies [23,24,25] have developed intelligent glass that rapidly switches between transparency and opacity using a hydrated PNIPAM film. The glass is transparent below 32.5 °C and becomes opaque above this temperature, with the switch between transparency and opacity induced by light or direct thermal stimuli. Additionally, the application of CST polymers can improve existing technologies. For example, they are expected to enhance the performance of DNA biosensors by allowing precise DNA orientation in PNIPAM-DNA conjugates when PNIPAM is in its collapsed state (Figure 1c) [26].

The phenomenon of CST-based transitions arises from the formation of dynamic hydrogen bonds and hydrophobic interactions that behave differently at various temperatures. Although these polymer systems have been extensively studied over the past few decades and are now partially commercially available, the molecular mechanisms driving temperature-induced transitions have not yet been explored adequately in many cases. To the best of our knowledge, no comprehensive effort has been made to systematically examine the role of different types of dynamic hydrogen bonds and hydrophobic interactions across various polymer systems. The dynamic nature of hydrogen bonds allows them to react to thermal energy and water molecules but also adjust to changes in the polymer structure. Furthermore, the role of the hydration shell around hydrophobic polymer segments is rarely addressed in the literature. Given significant advancements in temperature-responsive polymer systems exhibiting CST-based transitions, the objective of this work is to consolidate current knowledge on dynamic hydrogen bonds and hydrophobic interactions at T < LCST and T > LCST.

## 2. Phenomenon of Critical Solution Temperature

The behavior of the polymer in solution depends on the hydrophobic-hydrophilic balance of the polymer and the properties of the solvent. This is a fine line where the smallest functional group in the polymer chain can determine whether the polymer is soluble, insoluble, or capable of temperature-induced transitions. The presence of a CST in polymers is governed by the interactions between the polymer and solvent, and the thermodynamic parameters that dictate their mixing. As was mentioned before, there are two primary types of CSTs: LCST and UCST.

Figure 2 presents the phase diagrams of thermoresponsive polymer solutions that exhibit LCST or UCST behaviors, highlighting the types of interactions involved in different polymer phase separation processes: (a) hydrophobic interactions in P*N*IPAM, (b) hydrogen bonding interactions in poly(*N*-acryloyl glycinamide) (P*N*AGA), and (c) ionic interactions in polysulfobetaines (PSB) [27]. For P*N*IPAM, the formation of hydrogen bonds between polymer segments also plays an important role above the LCST transition.

The entropy-driven LCST thermoresponse observed for nonionic P*N*IPAM (Figure 2a) is related to the hydrophobic effect [28]: Ordered hydration structures of the water cages surrounding the repeating units result in a negative entropy of mixing Δ*S*_m_. The polymer dissolves, below the LCST temperature, as long as the Gibbs energy of mixing, Δ*G*_m_ = Δ*H*_m_ − TΔ*S*_m_, is negative. This is secured by hydrogen bonds between polymer segments and water, which yield negative values for the enthalpy of mixing Δ*H*_m_. However, at higher temperatures, the hydrophobic effect dominates: the hydration shells are destroyed, and the segment–segment interactions (intramolecular and intermolecular) are favored, leading to a significant increase in the entropy Δ*S*_m_ and positive values for Δ*H*_m_, respectively. Phase separation and the coil-to-globule transition are observed. The hydrophobic effect is enhanced by the structure of the amphiphilic polymer, with hydrophobic and hydrophilic groups present in P*N*IPAM [28]. Furthermore, hydrogen bonds between PNIPAM amide fragments (not marked in Figure 2a and omitted in some earlier work [27]) can contribute to intra- and inter-chain segment–segment interactions.

In turn, the enthalpy driven UCST transition (Figure 2b,c) is caused by polymer-polymer interactions such as hydrogen bonding between H donors and acceptors of P*N*AGA (Figure 2b) or electrostactic forces between oppositively charged ionic groups of PBS (Figure 2c). Resulting positive enthalpy of mixing Δ*H*_m_ promotes phase separation at low temperatures. As the temperature increases for positive Δ*S*_m_, the entropic contribution (-TΔ*S*_m_) to the Gibbs enery of mixing Δ*G*_m_ becomes substantial, and above UCST the polymers dissolve in a solvent. At the CST, dissolution and phase separation are in equilibrium, meaning Δ*G*_m_ equals zero. Thus, CST can be expressed as CST = Δ*H*_m_/Δ*S*_m_. Increasing CST can be achieved by raising the Δ*H*_m_/Δ*S*_m_ ratio, which can be accomplished by either increasing Δ*H*_m_ or decreasing Δ*S*_m_.

Several additional factors influence CST in polymer systems. A key factor is the molecular weight of temperature-responsive polymers; higher molecular weights tend to shift the CST, altering the temperature at which phase separation occurs [2,3]. Polymer concentration is another important factor, as often concentrated and diluted solutions can exhibit distinct phase behaviors. The type of solvent and the presence of impurities also play a role in determining whether the solvent is good or poor for the polymer, influencing whether the system shows LCST or UCST behavior [2,3]. Lastly, the composition of copolymers or polymer systems, including the ratio and nature of different monomers, affects the presence and position of CST due to the varying solvent affinities [2,3].

In summary, the presence and type of CST in polymer solutions depend on the balance between enthalpic and entropic contributions to the mixing process, as well as factors such as polymer–solvent interactions, molecular characteristics, and environmental conditions. This understanding is essential for designing polymer systems tailored for specific applications, such as smart materials and drug delivery systems. All temperature-induced processes based on LCST or UCST are driven by interactions between specific functional groups or macromolecular fragments, either with each other or with water molecules. In the following subsections, we will explore the molecular mechanisms behind temperature-induced transitions in various temperature-sensitive polymer systems, including poly(acrylamide) and poly(*N*-alkyl acrylamide)s, polyacrylamide derivatives containing amino acid fragments, poly(methacrylamide) and poly(*N*-alkyl methacrylamide)s, poly(*N*-vinylalkylamides), lactam/pyrrolidone/pyrrolidine-based polymers, hydroxyl-containing polymers, and others.

## 3. Molecular Design and Mechanisms of the Temperature-Induced Transition of Homopolymers in Water

### 3.1. Molecular Design and Mechanism of the Temperature-Induced Transition in Poly(N-alkyl acrylamide)s Based Polymers

Poly(N-alkyl acrylamide)s are among the most widely studied thermoresponsive polymers, derived from the acrylamide family. In these polymers, an *N*-alkyl group with a variable alkyl chain is attached to the nitrogen atom in the acrylamide structure [29]. Their properties can be fine-tuned by adjusting the alkyl group, making them highly versatile. Figure 3 shows the temperature-responsive structures of this group. Due to their tunable solubility, thermal responsiveness, and biocompatibility, these polymers have found extensive applications in biomedical materials, drug delivery systems, and hydrogels [30]. Interestingly, LCST-like behavior has been reported in polyacrylamide (PAAm) in at least two studies [31,32]. In work [31] temperature-responsive properties of PAAm-grafted brush coatings were revealed, exhibiting an LCST-like transition at approximately 11 °C. These coatings showed only slight changes in thickness and surface morphology, but significant alterations in wettability. Hydrogels based on AAm, cross-linked with *N,N*′-methylene bis(acrylamide), also exhibited LCST behavior near human body temperature [32].

At room temperature, that is, at T > LCST, PAAm in water forms various hydrogen bond structures, including free amide groups (Figure 4a), *cis-trans*-multimers (Figure 4b) and *trans*-multimers (Figure 4c) of amide groups [33,34,35,36,37]. However, the proportion of *cis-trans* and *trans*-associates is much smaller than that of the free amide groups. In work [31], a thermal response mechanism was proposed for PAAm that is distinct from the ‘classical’ LCST transitions seen in polymers such as P*N*IPAM and POEGMA. This behavior is mainly attributed to hydrogen bond conformations between the hydrophilic amide groups of PAAm and water (Figure 4a), which predominate at lower temperatures. At T > LCST, the bonding changes, leading to increased hydrogen bonds among the amide groups within the PAAm chains (Figure 4b,c), mimicking the LCST behavior. Temperature-induced dehydration of free amide groups (Figure 4a) is believed to promote hydrogen bonding between amide fragments (Figure 4b,c), although these remain partially hydrated. The overall balance between water-bonded amide groups and inter-amide hydrogen bonds changes only slightly at the transition, with a minimal change in the total number of water molecules bound to amide groups [31].

Table 1 presents LCST data for *N*-substituted polyacrylamides from various sources [38,39]. The listed monomers include both *N*-monosubstituted and *N,N*-disubstituted acrylamides. The thermosensitivity and solution behavior of these polymers depend on the *N*-alkyl substitution in the acrylamide monomers. For instance, acrylamides with one or two methyl groups, such as poly(*N,N*-dimethylacrylamide), are fully water soluble, whereas those with more hydrophobic *N*-alkyl groups, such as n-butyl, iso-butyl, sec-butyl, and tert-butyl, are insoluble in water. Similarly, acrylamides with two propyl groups or a combination of propyl and ethyl groups also exhibit water insolubility.

Despite the wide range of thermoresponsive poly(*N*-alkyl acrylamide)s, only poly(*N*-isopropylacrylamide) (P*N*IPAM) and its copolymers have practical applications. Surface-grafted P*N*IPAM brushes have received significant attention in literature. The LCST of P*N*IPAM and its copolymers is around 32 °C, which is close to the physiological temperature. P*N*IPAM is widely used in biomedical applications such as biosensors [40], thermally modulated drug and gene delivery systems [41,42], and P*N*IPAM-conjugated proteins for thermally regulated enzyme function [43].

The temperature-responsive behavior of poly(*N*-monoalkylacrylamide)s with LCST, especially P*N*IPAM, was mainly attributed to hydrogen bonds between the hydrophilic amide groups of the P*N*IPAM segments and water, which are dominant below LCST but are replaced by hydrogen bonds between the amide groups in the P*N*IPAM chains above LCST [44,45] (see Figure 5a). At temperatures below LCST, the water molecules create a hydrophilic layer around the hydrophobic groups, which is destroyed at LCST, and the hydrophobic interactions begin to play a key role. This induces the transition of P*N*IPAM from hydrated loose coils to hydrophobic collapsed chains.

A different temperature-induced transition mechanism occurs in dialkyl-substituted polyacrylamides, as detailed for P*N*DEAM [46]. At T < LCST, the carbonyl groups in P*N*DEAM are associated with water molecules through hydrogen bonds, and the ethyl groups are hydrated. In turn, at T > LCST, the ethyl groups become dehydrated and engage in hydrophobic interactions. However, the carbonyl groups remain partially hydrated and continue to form hydrogen bonds with water molecules after the phase transition. The heat of transition (ΔH) mainly reflects the breaking of hydrogen bonds within the structured hydration shell surrounding the ethyl groups.

### 3.2. Molecular Design and Mechanism of the Temperature-Induced Transition in Other N-Subtituted Deravatives of Polyacrylamide Based Polymers

Another intriguing class of temperature-responsive polymers is based on acrylamide derivatives of amino acids. Instead of the alkyl motifs, these polymers include the fragments of amino acids. Figure 6 shows the structures of the temperature-responsive derivatives of polyacrylamide containing amino acid fragments.

Poly(acryloyl-L-proline methyl ester) (PAProM), which includes both amide and ester groups in its monomer units, has drawn particular interest. PAProM exhibits LCST behavior between 14–18 °C [47,48], with hydrogels that show LCST around 20 °C [49]. These hydrogels have potential in drug-controlled release systems [50,51]. Another polymer with a similar chemical structure and temperature-responsive properties is poly(*N*-acryloyl-4-*trans*-hydroxy-L-proline methyl ester) (poly(A-Hyp-OMe)). Unlike PAProM, poly(A-Hyp-OMe) undergoes a clear phase transition after heating to a higher temperature (49.5 °C), suggesting that the addition of a hydroxyl group to the monomer unit significantly increases the transition temperature [48].

The LCST mechanism for PAProM has been extensively studied using infrared spectroscopy [52]. PAProM has no H-bond donors because intrachain H-bonds was ruled out. Most water molecules form hydrogen bonds with the amide or ester carbonyls of PAProM. At T < LCST, nearly all ester carbonyls of PAProM accept a single hydrogen bond from a water molecule. Additionally, the average number of hydrogen bonds to the ester carbonyls is smaller than those to the amide carbonyls across all temperatures. Two types of interactions between water and alkyl fragments were proposed: direct and indirect. In direct interaction, the alkyl group and water form a weak hydrogen bond (C-H·····OH_2_), while in indirect interaction, adjacent hydrogen bonds formed by polar functional groups influence the hydration (referred to as hydrophobic hydration). Given the limited amount of adsorbed water under these conditions, most the water molecules bind to the amide or ester carbonyls of PAProM. Some dehydration of the polymer chain occurs upon phase separation. During this phase change, part of the singly hydrogen-bonded carbonyls break, forming free carbonyls. At 50 °C, the ratio of free carbonyls is estimated to be 16%. Phase separation involves breaking hydrogen bonds to both ester and amide carbonyls and dehydrating alkyl groups on the main and side chains. Below the LCST, 63% of the amide carbonyl population binds to two water molecules, while 33% binds to one water molecule. Above the LCST, the proportion of amide carbonyls bound to two water molecules decreases. For the ester carbonyls, 100% bind to one water molecule below the LCST, and 84% remain bound above the LCST, with the rest being free. This indicates that ester carbonyls form fewer hydrogen bonds than amide carbonyls, and some break during phase separation.

It has been sixty years since the monomer *N*-acryloyl glycinamide (*N*AGA) and its polymer poly(*N*-acryloylglycinamide) (P*N*AGA) were first reported in 1964. Over the past decades, significant advances have been made in understanding and utilizing P*N*AGA aqueous solution systems across a broad concentration range. These systems have exhibited distinct thermoresponsive behaviors and performance variations, making P*N*AGA an intriguing polymer in the field of temperature-responsive materials. Studies on P*N*AGA suggest that its UCST behavior is influenced by factors such as polymer concentration, molecular weight, and the presence of salts or cosolvents in solution. Studies on P*N*AGA suggest that its UCST behavior is influenced by factors such as polymer concentration, molecular weight, and the presence of salts or cosolvents in solution. P*N*AGA is the most well-known polymer with UCST [53,54,55,56], the hydrogen bonds between the carbonyl and amine groups of P*N*AGA were observed at T < UCST. They broke down at T > UCST when the interaction with water prevailed for both groups (Figure 7) [15].

### 3.3. Molecular Design and Mechanism of the Temperature-Induced Transition in Poly(N-alkyl methacrylamide)s Based Polymers

Poly(*N*-alkyl methacrylamide)s are polymers, which are structurally similar to their acrylamide counterparts but have a methyl group (-CH_3_) attached to the backbone, providing different thermal and solubility properties. Examples of poly(*N*-alkyl methacrylamide) that are cable of temperature-induced transitions are described in Figure 8. The effects of the poly(*n*-alkyl methacrylamide) substitution groups on the LCST or solubility are summarized in Table 2 [39].

The progenitor of this class, poly(methacrylamide) (PMAAm), exhibits temperature-responsive behavior due to its ability to form intra- and intermolecular hydrogen bonds (Figure 9). Early studies did not observe phase separation due to the use of ionic initiators. However, PMAAm shows UCST behavior when synthesized with a nonionic initiator, with phase separation occurring upon cooling around 40.5 °C and during heating at 57 °C, although the transition is slow, leading to hysteresis. The additional methyl group makes PMAAm more hydrophobic than polyacrylamide, contributing to its thermal behavior [57].

Some poly(*N*-alkyl methacrylamide)s exhibit LCST behavior, with mechanisms likely similar to those of poly(*N*-alkyl acrylamide)s. However, these polymers generally exhibit higher LCSTs than their acrylamide analogs. For example, poly(*N*-isopropylmethacrylamide) (P*N*IPMAM) has an LCST of around 45 °C in water [58], attributed to steric hindrance from the additional methyl group, which prevents hydrophobic side groups from interacting, resulting in a higher transition temperature of 38–42 °C [59,60]. During the phase transition, P*N*IPMAM forms less stable aggregates that are almost completely dehydrated above the LCST, unlike P*N*IPAM [61]. In similar work [62], it was shown that compared to P*N*IPAM, P*N*IPMAM exhibits similar hydration capacity but a higher phase transition temperature, increasing from 29 °C to 38 °C. This suggests that while the extra methyl group does not significantly affect the overall hydration, it restricts conformational flexibility during the thermal transition.

### 3.4. Molecular Design and Mechanism of the Temperature-Induced Transition in Poly(N-vinylalkylamide) Based Polymers

Poly(*N*-vinylalkylamide)s, structural analogues of poly(*N*-alkyl acrylamide)s, have attracted attention due to their temperature-responsive properties. The chemical structures of these temperature-responsive poly(*N*-vinylalkylamide)s are shown in Figure 10.

In work [63] a series of poly(*N*-vinylalkylamide) derivatives with varying alkyl side chain lengths were synthesized. These polymers feature amide side chains where nitrogen is directly bonded to the polymer backbone. Poly(*N*-vinylisobutyramide) (P*N*VIBA), a structural isomer of P*N*IPAM, shows a sharp thermal transition at 39 °C, with the LCST strongly dependent on polymer concentration [63]. Similar findings have been reported, that do not show a molecular weight effect on LCST [64]. Although the hydrophilic-hydrophobic balance of P*N*VIBA is similar to that of P*N*IPAM, its higher transition temperature may be due to the microstructure of the hydrated polymer. Recent studies [65] have shown that P*N*VIBA exhibits less swelling than P*N*IPAM, has a lower water affinity, and forms larger hydrophobic clathrates, which inhibit water penetration. Moreover, nitrogen-dried P*N*IPAM films retain primary water, whereas P*N*VIBA can be completely dried. The initial stage of water uptake in the main polymer layer involves filling the free volume, resulting in a water content of 3.8% in P*N*VIBA compared to 6% in P*N*IPAM. Other derivatives, such as poly(*N*-vinyl-*n*-butyramide) (P*N*VBA), demonstrate a sharp thermal transition at 32 °C [66]. Poly(*N*-vinylalkylamide)s are likely to exhibit a temperature-induced transition mechanism similar to that of poly(*N*-alkyl acrylamide)s. However, the impact of the hydrated structure of the polymer contributes to an increase in LCST.

### 3.5. Molecular Design and Mechanism of the Temperature-Induced Transition in Lactam-/Pyrrolidone-/Pyrrolidine-Based Polymers

Lactam-, pyrrolidone-, and pyrrolidine-based polymers incorporate nitrogen-containing heterocyclic compounds, contributing unique properties because of the nitrogen atom and ring structure. Representatives of this class that respond to temperature are shown in Figure 11.

Poly(*N*-vinyl caprolactam) (PVCa), for example, is a non-ionic, water-soluble, biocompatible polymer with an LCST around 31 °C, which is nearly independent of the concentration of PVCa in dilute solutions [67,68]. On the contrary, poly(*N*-vinylpyrrolidone) (PVPy) only exhibits LCST behavior in high salt environments, such as 1.5 M potassium fluoride solutions, where it separates into phases at about 30 °C [69]. Another polymer, poly[*N*-(2-methacryloyloxyethyl) pyrrolidone] (P*N*MP), shows a sharp phase separation at LCST equal to 52 °C [70], and poly(3-ethyl-*N*-vinyl-2-pyrrolidone) (C2PVP) shows phase separation at LCST above 26 °C [71]. Poly(*N*-ethylpyrrolidine methacrylate) (PEPyM) has LCST at 15 °C, while poly(*N*-acryloylpyrrolidine) (PAPR) exhibits LCST at 51 °C [72,73].

The thermoresponsive behavior of a polymer based on a pyrrolidone structure in aqueous solution is shown in Figure 12 [71]. We expect a similar behavior across the polymer group presented here. At T < LCST, hydrophilic interactions between polymer and solvent molecules dominate, maintaining the water solubility of the polymer. As illustrated in Figure 12 [71], the solubility of poly(3-ethyl-*N*-vinyl-2-pyrrolidone) at T < LCST is also aided by hydrophobic hydration around the carbon atoms of the side-chain, where cage-like water formations stabilize the polymer in solution. At the same time, water molecules interact with the oxygen atoms in lactam rings. However, when the temperature exceeds the LCST, hydrophobic interactions within the polymer chains become dominant, causing a sharp phase transition and polymer phase separation. In this case, weak “cross-linking” structures formed by hydrogen bonds between one water molecule and two oxygen atoms of adjacent lactam rings are observed. At T > LCST, the cage-like water formations break down, releasing a significant amount of water molecules. The dehydration process follows the order: ethyl groups > C=O groups > CH_2_ groups in the ring, with the reverse sequence observed during rehydration. A similar LCST mechanism has been described for poly(*N*-vinylcaprolactam) [69].

### 3.6. Molecular Design and Mechanism of the Temperature-Induced Transition in Poly(oligo(ethylene glycol) methacrylate)s—(POEGMA)s

Among the various ethylene glycol-based macromonomers, oligo(ethylene glycol) methacrylates (OEGMA) have been the most extensively studied. Poly(oligo(ethylene glycol) methacrylate) (POEGMA) are group polymers with similar chemical structures (Figure 13). They are biocompatible, nonionic, water-soluble, non-toxic, and non-immunogenic, making them some of the most commonly used synthetic polymers in biomedical applications [74,75]. The LCST of POEGMA, as well as its antifouling and stealth behavior [76,77,78], strongly depends on the number of ethylene glycol units and the presence of methyl or ethyl ether groups in the monomer (Table 3) [77,79].

For example, POEG_4_MEMA has an LCST of 68 °C, significantly higher than POEG_2_MEMA, and POEG_3_MEMA, which have LCST values of 26 °C and 52 °C, respectively (see Figure 14). This illustrates the impact of the number of ethylene glycol units on LCST. In contrast, POEG_2_EEMA is only soluble in cold water below 4 °C. The LCSTs for POEG_2_EEMA, POEG_3_EEMA, and POEG_3_EEMA were 4 °C, 27 °C, and 42 °C, respectively, showing a continuous rise with increasing side-chain length. In particular, the LCST values for the ethyl ether derivatives were 22–26 °C lower than their methyl counterparts, confirming that the ethyl group is more hydrophobic. This hydrophobicity likely hinders side-chain hydration, lowering the LCST compared to those of the methyl ethers. Among homopolymeric POEGMAs, only a few, such as POEG_2_MEMA and POEG_3_EEMA, with LCST near physiological temperature (26 and 24 °C, respectively) have practical applications [77].

Figure 15 shows a hypothetical scenario that illustrates hydrogen bonding between POEGMA ether and carbonyl oxygens and water molecules and van der Waals interactions, present at various temperatures in POEGMA246 coatings. At T < LCST, hydrogen bonding between the POEGMA ether and carbonyl oxygens and water hydrogens dominates. Hydration around the side chains maintains solubility in water at low temperatures, but this balance changes at T > LCST, where polymer–polymer interactions become thermodynamically favorable, altering the structure of the water and releasing water molecules [80,81]. Noteworthy results were obtained in [82], where ellipsometry measured the water content in POEGMA246 to be 91.6% at T < LCST and 87.1% at T > LCST. Interesting results were presented in [83,84], where it was shown that for POEGMAs, hydrogen bonds form exclusively between the carbonyl oxygens and the hydrogen atoms of water at temperatures below the LCST. Above the LCST, most of these hydrogen bonds are disrupted; however, a certain amount of bound water remains within the polymer. This suggests that the hydrophilic layer of water surrounding the polymer fragments likely plays a key role in retaining water. Various proposed schemes describing different hydrogen bonding conformations highlight the need for further studies to clarify the exact nature and dynamics of these interactions. In [62], the hydration and dehydration kinetics of P*N*IPMAM and POEG_2_MEA films were compared. These films exhibit markedly different behaviors under thermal stimuli. PNIPMAM demonstrates a higher hydration capacity due to the presence of N–H and C=O groups, but it hydrates more slowly owing to its higher glass transition temperature. Upon heating, both films undergo shrinkage, molecular rearrangement, and partial reswelling, with P*N*IPMAM showing a faster and more pronounced response. In contrast, POEG_2_MEA lacks N–H groups, resulting in lower hydration capacity and distinct interfacial behavior.

### 3.7. Molecular Design and Mechanism of the Temperature-Induced Transition in Poly(oligo(ethylene glycol) acrylate)s—(POEGA)s

The less hydrophobic backbone of poly(oligo(ethylene glycol) acrylate)s (POEGA)s compared to those of POEGMAs opens up new possibilities for thermosensitive monomers, including shorter ethylene glycol chains. Despite their advantages, POEGAs are less well-represented in the literature. The chemical structures of the POEGAs that respond to temperature are presented in Figure 16 and the properties in Table 4. POEG_1_MEA, the most hydrophobic of POEGAs, has the lowest reported LCST at 5 °C [85,86], and for other POEGAs, LCST ranges from 10 to 92 °C depending on their structure (number of ethylene glycol units and structure of methyl or ethyl ether), synthesis methods and LCST measurement conditions [83,87,88,89,90]. This suggests that LCSTs generally increase with side-chain length and are higher for methyl ethers compared to ethyl ethers.

POEG_1_MEA, the most hydrophobic of POEGAs, has the lowest LCST, with reported values of 5 °C for the homopolymer and nearly 0 °C when synthesized by a different method [85,86,87]. POEG_2_MEA has the widest range of reported LCSTs among POEGAs. Depending on the synthetic method, the weight of the polymer, and the concentration, its LCSTs range from 9 to 45 °C [83,87,88,89,90]. In most cases, the LCST of POEG_2_MEA is approximately 40 °C. POEG_3_MEA demonstrated LCSTs ranging from 75 °C at 0.05 wt% to 56 °C at 1 wt%, showing a similar inverse relationship between concentration and LCSTs, with this effect diminishing beyond 1 wt%, as previously discussed for POEG_2_MEA. A similar dependency on molecular weight was observed where LCST increased from 46 to 61 °C (0.5 wt%) as the molecular weight almost doubled [87]. Finally, POEG_8.5_MEA, that is, 8.5 egus on average (8–9) can be considered the most hydrophilic monomer in that class of polymers that still has a reported LCST for homopolymer with LCST of almost 92 °C [91]. The thermoresponsive properties of POEG_2_EEA were first reported by Lutz et al., this polymer showed an LCST range of 10 to 16.5 °C according to the synthesis methodology [82,91,92,93,94]. Only a few scientific publications reported the thermoresponsive POEG_3_EEA, with LCST in the range of 34 to 39 °C depending on the synthesis methodology [95,96]. These results have not shown any noticeable impact of the molecular weight on LCST. In general, we see a similar tendency to POEGMAs, the increase of the values of the LCSTs with the increase of the number of ethylene glycol units, and the strong increase of the LCSTs for methyl ethers compared to that for ethyl ethers.

The mechanism of temperature-induced transitions based on LCST for POEGAs, as well as POEGMAs, was investigated using POEG_2_MEA and POEG_2_MEMA as examples [83]. POEG_2_MEA and POEG_2_MEMA exhibited LCST at 45 °C and 26 °C, with transition enthalpies of 21 J/g and 36 J/g, respectively. The behavior of both polymers during phase separation was quite similar, with the exception of hydrogen bonding to the carbonyl groups. Although free carbonyl groups existed in both polymers even at temperatures below the LCST and at low polymer concentrations, POEG_2_MEA exhibited a higher average number of hydrogen bonds compared to POEG_2_MEMA. The additional methyl groups in POEG_2_MEMA likely hinder the formation of hydrogen bonds due to steric effects. Consequently, increased hydrophobicity and steric hindrance of the methyl groups make the aqueous solutions of POEG_2_MEMA less stable than those of POEG2MEA.

### 3.8. Molecular Design and Mechanism of the Temperature-Induced Transition in Hydroxyl-Containing Polymers

Hydroxyl-containing polymers are a class of polymers that have incorporated hydroxyl groups within their structure. Unlike other temperature-sensitive polymers, their use is less common, with only a few examples reported. Figure 17 illustrates hydroxyl-containing polymers capable of temperature-induced transitions. A lesser-known fact is that the well-known homopolymer poly(2-hydroxyethyl methacrylate) (PHEMA), often considered water-soluble, can exhibit temperature responsiveness with LCST under specific conditions. As suggested by S. Armes and co-workers, PHEMA is typically described as only water-swellable. However, low-molecular-weight PHEMA (with a degree of polymerization below 20) is water-soluble over a wide temperature range without exhibiting LCST. In contrast, PHEMA homopolymers with a degree of polymerization between 20 and 45 showed inverse temperature solubility in dilute aqueous solutions at pH 6.5, with the LCST systematically increasing as the degree of polymerization decreased. At degrees of polymerization above 50, PHEMA becomes insoluble [97].

Poly(2-hydroxypropyl acrylate) (PHPA) is perhaps the most unique polymer with an LCST, resulting in a relatively hydrophobic polymer despite containing a hydroxyl group. Interestingly, PHPA is sold and used as a mixture of isomers, making each homopolymer essentially a statistical copolymer. The isomeric mixture comprises approximately 75% 2-hydroxypropyl acrylate (2-HPA, Figure 13) and 25% 1-methyl-2-hydroxyethyl acrylate (1-MeHPA), produced by opening the ring of propylene oxide with acrylic acid [98]. The first report on the thermoresponsive properties of PHPA was published by Taylor and Cerankowski in 1975 during their study on new polymer films with temperature-dependent permeability [99]. They reported an LCST of 16 °C (10 wt%), which remains the lowest LCST reported for PHPA in the scientific literature. Schubert et al. later investigated the thermoresponsive properties of PHPA, showing that polymer concentration strongly affects LCST [100] with LCST that ranges from 18.3 °C at 1.5 wt% to 33.3 °C at 0.25 wt%. In other work [101] homopolymers and block copolymers of poly(2-hydroxypropyl acrylate), which exhibit LCST behavior between 30 °C and 60 °C, were described.

Poly(pentaerythritolmonomethacrylate) (PPM), with three free hydroxyl groups, has been studied for its grafted polymer brushes, which exhibit temperature-induced transitions around 14 °C [102]. These studies also showed that partial post-polymerization modification of hydroxyl groups (up to a few percent) was possible in grafted PPM brushes using acetyl chloride and pyromellitic acid chloride without compromising the thermal response of the coatings. In a related study [103], two new glycerol ether-based poly(meth)acrylates with β-hydroxy-functional side chains were synthesized: structurally isomeric poly(3-ethoxy-2-hydroxypropyl) acrylate (PEHPA) and poly(2-hydroxy-3-methoxypropyl methacrylate) (PHMPMA). The distinct amphiphilic balance of PEHPA, due to its higher side-chain hydrophobicity, resulted in lower LCST (22–33 °C). In contrast, the increased hydrophobicity of the backbone in PHMPMA led to higher LCST (37–67 °C) and greater sensitivity to both intrinsic and extrinsic factors.

The mechanism of temperature-induced transitions in water for hydroxyl-containing polymers with LCST is likely similar to that described for PHEMA in mixed organic solvents [104]. A more detailed explanation of the behavior of LCST in poly(hydroxypropyl acrylate) was provided in [105] and is depicted in Figure 18. The interactions of hydroxyl groups in PHPA hydrogels can be classified into three main types: (1) interactions between polymer hydroxyl groups and water molecules (–OH······OH_2_), (2) interactions between hydroxyl groups within polymer chains (–OH······OH–), forming intramolecular hydrogen bonds, and (3) interactions between hydroxyl groups of water molecules (H_2_O······OH_2_). At T < LCST –OH······OH_2_ hydrogen conformations predominate, suggesting better polymer-water compatibility. As the temperature increases, the proportion of –OH······OH_2_ bonds decreases, and –OH······OH– bonds increase, indicating the disruption of polymer-water interactions and the formation of new hydrogen bonds between polymer chains. Additionally, increasing temperatures promote the aggregation of side chains.

There were also two types of carbonyl structures: one was due to hydrogen bond formation with the polymer hydroxyl group (–C=O·····HO–); and another was a hydrated carbonyl group (–C=O·····H_2_O). As the temperature increased, the percentage of of –C=O·····H_2_O bonds decreased, while the percentage of -C=O······HO– bonds increased. In general, the hypothetical conformations of the van der Waals interactions and hydrogen bonding among the hydroxyl groups of the hydroxyl containing polymers and the water molecules at various temperatures are presented in Figure 18. At temperatures below LCST the hydroxyl groups of the polymers interact with molecules of the water forming the hydrogen bonds. In contrast, at temperatures above LCST, hydrogen in the hydroxyl groups of the polymer interacts with oxygen in the other hydroxyl group and, with hydrophobic-hydrophobic interactions, plays a key role in the polymers. Without doubts, the destruction of the hydrophilic layer around the hydrophobic fragments plays an important role in the LCST mechanism.

### 3.9. Molecular Design and Mechanism of the Temperature-Induced Transition in Polyvinylpyridines

Polyvinylpyridines (PVPy) are a class of synthetic polymers derived from vinylpyridine monomers, where a pyridine ring is attached to a vinyl group. They are characterized by their basicity and the ability to form coordination complexes because of the nitrogen atom in the pyridine ring. One of the PVPy is poly(4-vinylpyridine) (P4VP) presented in Figure 19 is a hydrophobic polymer that is insoluble in water until more than ca. 35% of the pyridine groups are charged, e.g., by protonation [106]. However, pyridine molecules can form hydrogen bonds with water molecules [107,108]. Moreover, pyridyl groups were assumed to respond to moisture adsorption in shape memory polyurethanes containing pyridine moieties, as synthesized by Chen’s group [109,110].

Although the pH-responsive properties of polyvinylpyridines are well-known, the impact of temperature on the properties of this polymer was demonstrated only in a small amount of works [111,112], where poly(4-vinylpyridine) grafted brush coatings were synthesized and their behavior mimicking LCST was demonstrated. P4VP coatings exhibit a temperature dependence of the water contact angle with a well-defined transition at 13–14 °C. This transition is absent at acid pH levels wherein almost all pyridyl groups are protonated.

The hypothetical scenario of van der Waals interactions and hydrogen bonding between PVP pyridyl groups, PVP protonated pyridyl groups, and water molecules at various pH and temperatures for P4VP coatings is described in Figure 20. For neutral and high pH values, the thermal response is mainly attributed to the hydrogen bonds between the nitrogen from P4VP and water hydrogen. In turn, at T > LCST, polymer–polymer interactions (van der Waals interactions) are thermodynamically more favored than polymer–water interactions, inducing the transition of P4VP from a relatively hydrated state to a hydrophobic state. In addition to the temperature dependence, the P4VP grafted brush coatings also show a strong response to pH, similar to the high pH-sensitivity of pure P4VP chains. At low pH, the pyridyl groups are transformed into protonated pyridyl groups, which fosters repulsion between the positively charged pyridyl motifs in P4VP macromolecules. Moreover, the creation of hydrogen bonds with the oxygen of the water is strongly facilitated here. In works [113,114] where temperature and pH-responsive P4VP grafted brushes were used as templates for synthesis of silver nanoparticles, their thermo-switchable antibacterial activity was demonstrated.

### 3.10. Molecular Design and Mechanism of the Temperature-Induced Transition in Poly(methacrylic acid)

Poly(methacrylic acid) (PMAA) (Figure 21) is a polymer derived from the polymerization of methacrylic acid, a monomer containing both a carboxyl group (-COOH) and a vinyl group (-CH=CH_2_). PMMA is an anionic polymer that is important because of its pH-sensitive behavior, which makes it useful in a variety of applications, particularly in biomedical and drug delivery systems. Poly(methacrylic acid) (PMAA) has well-known pH-responsive properties and much less well-known temperature-responsive behavior [115,116,117]. In the work [116], the LCST for PMMA solutions was shown to be equal to 48 °C.

On the basis of literature sources, we propose an explanation of such behavior of PMAA coatings as related to the change in polymer-water and polymer-polymer interactions below and above the transition temperature. In Figure 22, the postulated interactions of carboxyl or carboxylate groups with water molecules at T < LCST, polymer-polymer links through doubly H-bonded dicarboxylic dimers, and van der Waals or hydrophobic interactions between methyl groups at T > LCST [116,118,119,120] are shown. Carboxylate groups (-COO^−^) interact strongly with water molecules through ion-dipole interactions. The partial positive charges on the hydrogen atoms of water molecules are strongly attracted to the negatively charged oxygen atoms of the carboxylate group. Lower amounts of carboxylate groups will favor temperature-induced transitions. Solvent-polymer interactions through H-bonds are energetically favorable to mixing, but entropically unfavorable, which causes the LCST behavior. At this time, polymer-polymer interactions through doubly H-bonded dicarboxylic dimers and van der Waals or hydrophobic interactions between methyl groups are preferred, and as a result, grafted PMAA molecules collapse. We assume that carboxylate groups also take part in these interactions. It is important to note that no temperature-induced transitions can be observed for polyacrylic acid, suggesting that the van der Waals forces due to methyl groups (-CH_3_) are crucial for the existence of LCST [119].

## 4. Conclusions

The temperature-induced transitions of polymer systems to temperature can be due to various mechanisms, of which CST is the best known. This is due to a number of breakthrough technologies, such as the fabrication of the tissue engineering platform [21,22] or the manufacturing of smart windows that reduce sunlight penetration during heating and thus create comfortable temperatures in the apartment [23,24,25]. Also, CST may be applied to improve existing technologies, such as controlled drug delivery [121] or to enhance the performance of DNA biosensors based on the precise orientation of DNA in P*N*IPAM-DNA conjugates at the onset of the collapsed state of P*N*IPAM [26]. On the basis of our previous experience, scientists understand that the thermosensitive properties are led by the CST phenomenon, but they do not go into the details of the dynamic hydrogen bonds and van der Waals interactions that enable it to be realized. It is very important to understand the molecular mechanisms that drive the CST-based phenomenon to understand how the temperature-sensitive properties of polymer systems can be changed. Additionally, understanding hydrogen bonding and van der Waals interactions in polymers with CST is crucial to predicting and controlling polymer behavior in various applications. Despite significant advances in the field of thermoresponsive polymer systems, many challenges and opportunities remain in the development of smart materials. In this review, we have compiled various examples of molecular mechanisms underlying CST-based phenomena, including both LCST and UCST transitions, which are driven by dynamic hydrogen bonding and van der Waals interactions in homopolymer systems containing a single type of functional group. We have extensively discussed the molecular mechanisms behind temperature-induced transitions in a range of temperature-sensitive polymer systems, such as poly(acrylamide) and poly(*N*-alkyl acrylamide)s, polyacrylamide derivatives with amino acid fragments, poly(methacrylamide) and poly(*N*-alkyl methacrylamide)s, poly(*N*-vinylalkylamides), lactam/pyrrolidone/pyrrolidine-based polymers, hydroxyl-containing polymers, and others. A deep understanding of the molecular interactions in homopolymer systems lays the foundation for further research into thermosensitive polymer systems with multiple functional units. Interestingly, the diversity of molecular mechanisms governing CST-based transitions and their continued study and application in smart materials are likely to yield many exciting discoveries in the future.

## Figures and Tables

**Figure 1 polymers-17-01580-f001:**
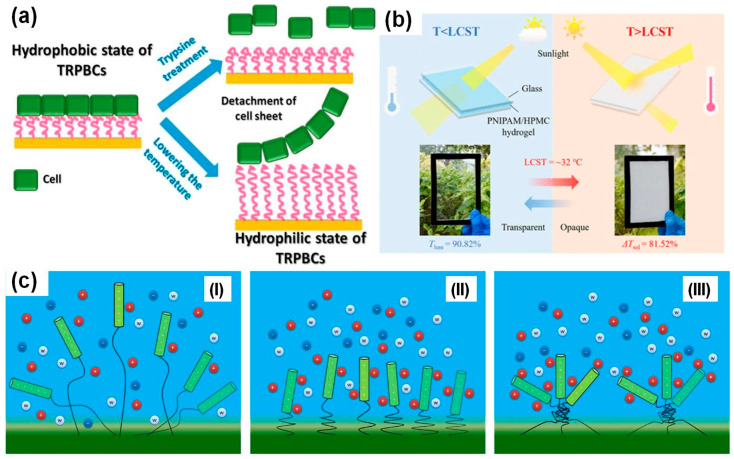
Cell growth (T > LCST) and detachment of the cell sheet from the grafted polymer brushes using trypsin (conventional methodology) or change of temperature (T < LCST) (**a**). Smart window with hydrated PNIPAM film exhibited high light transmittance at T < LCST and bed at T < LCST [23] (**b**). The orientation of DNA molecules (green) conjugated to PNIPAM chains at biosensor surfaces can be controlled with temperature (**c**) with the DNA order parameter: low at T < LCST when hydrophilic polymers extend randomly (**I**); high at the onset of LCST when polymers become hydrophobic and collapse sharply (**II**), and reduced at T > LCST when micro-phase separation of polymers appears [26] (**III**).

**Figure 2 polymers-17-01580-f002:**
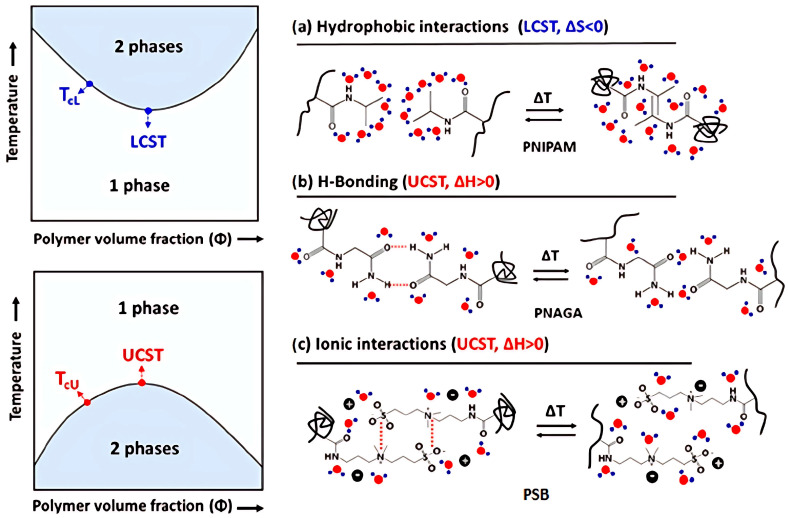
Phase diagrams of thermoresponsive polymer solutions with LCST or UCST. Types of interactions involved in different polymer phase separation processes: (**a**) hydrophobic interactions in P*N*IPAM, (**b**) H-bonding interactions in P*N*AGA, and (**c**) ionic interactions in PSB [27].

**Figure 3 polymers-17-01580-f003:**
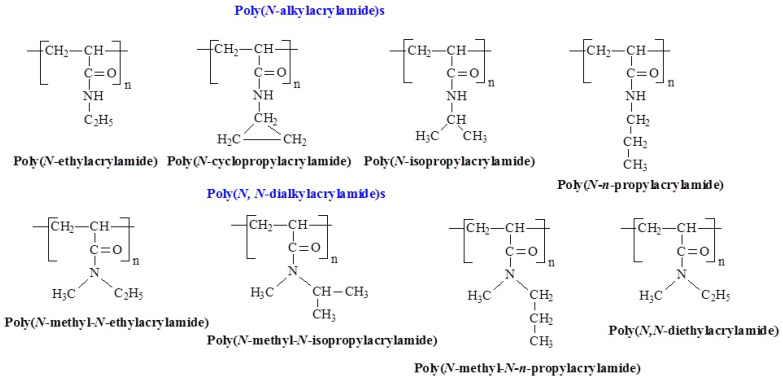
Chemical structures of temperature-responsive poly(*N*-alkyl acrylamide)s.

**Figure 4 polymers-17-01580-f004:**
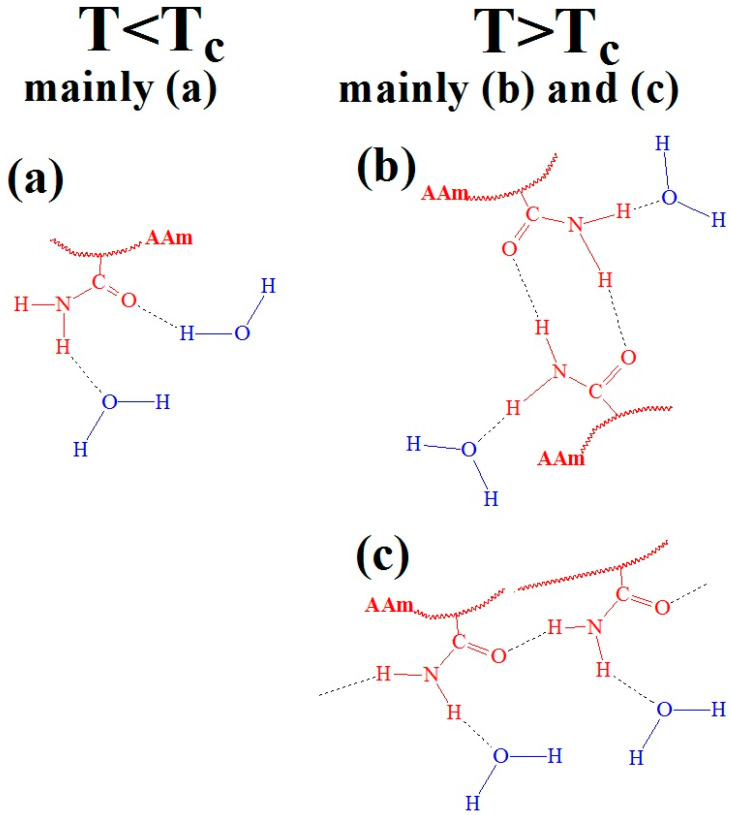
Hypothetical hydrogen bonding conformations between water and amide groups in PAAm, both above and below the LCST: (**a**) free amide groups, (**b**) *cis-trans*-multimers and (**c**) *trans*-multimers. Black letters indicate dominant conformations, while green letters represent minor conformations in different polymer states (T < T_C_ and T > T_C_).

**Figure 5 polymers-17-01580-f005:**
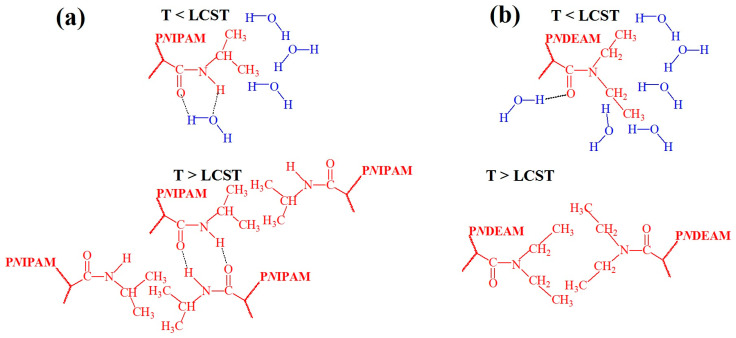
Hypothetical conformations in P*N*IPAM showing hydrogen bonding between amide groups and water molecules, along with the formation of a hydrophilic layer around hydrophobic fragments at T < LCST, and hydrogen bonding among amide groups coupled with hydrophobic-hydrophobic interactions between hydrophobic fragments at T > LCST (**a**). Hypothetical conformations in P*N*DEAM showing hydrogen bonding between oxygen atoms of amide groups and water molecules, along with the formation of a hydrophilic layer around hydrophobic fragments at T < LCST, and hydrophobic-hydrophobic interactions between hydrophobic fragments at T > LCST (**b**).

**Figure 6 polymers-17-01580-f006:**
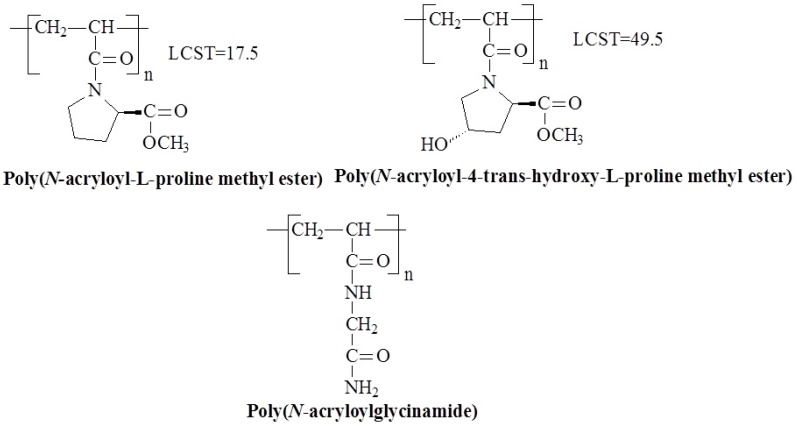
Chemical structures of temperature-responsive derivatives of polyacrylamide with amino acid fragments.

**Figure 7 polymers-17-01580-f007:**
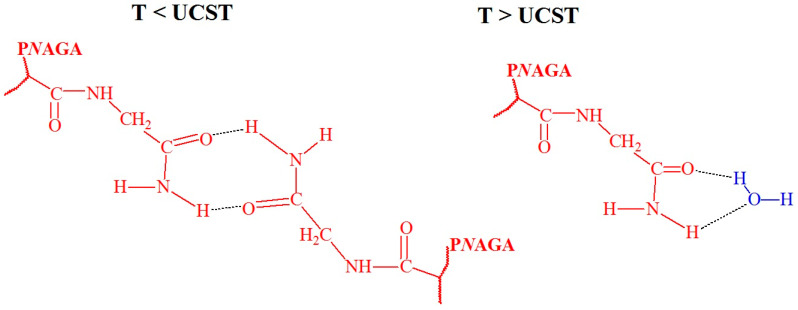
Hypothetical hydrogen bond conformations between the carbonyl and amine groups of P*N*AGA at T < UCST and water at T > UCST [15], no permission needed.

**Figure 8 polymers-17-01580-f008:**
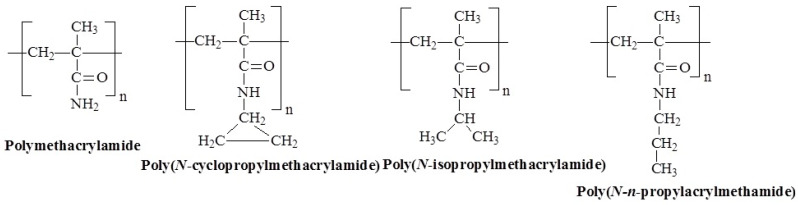
Chemical structures of temperature-responsive poly(*N*-alkyl methacrylamide)s.

**Figure 9 polymers-17-01580-f009:**
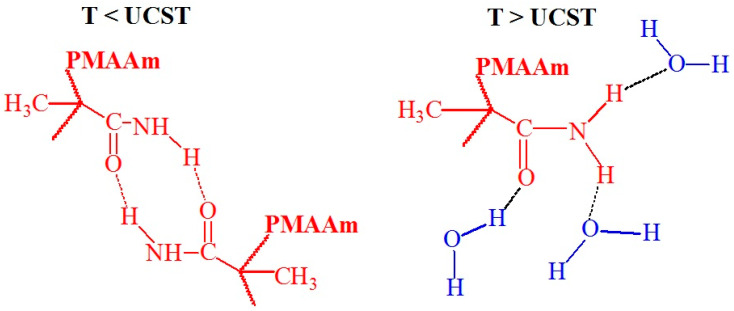
Hypothetical conformations of hydrogen bonding between the amide groups of PMAAm at T < UCST and water at T > UCST [15].

**Figure 10 polymers-17-01580-f010:**
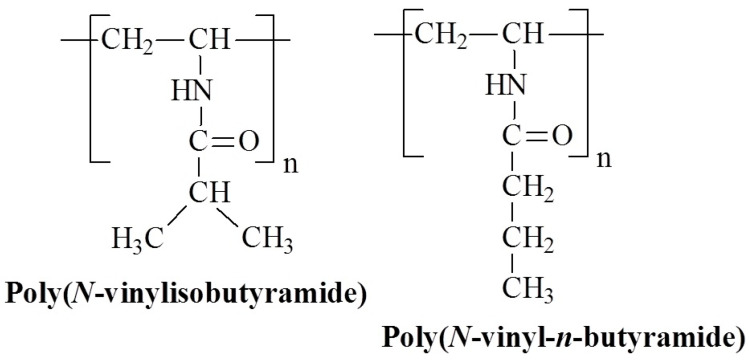
Chemical structures of temperature-responsive poly(*N*-vinylalkylamide)s.

**Figure 11 polymers-17-01580-f011:**
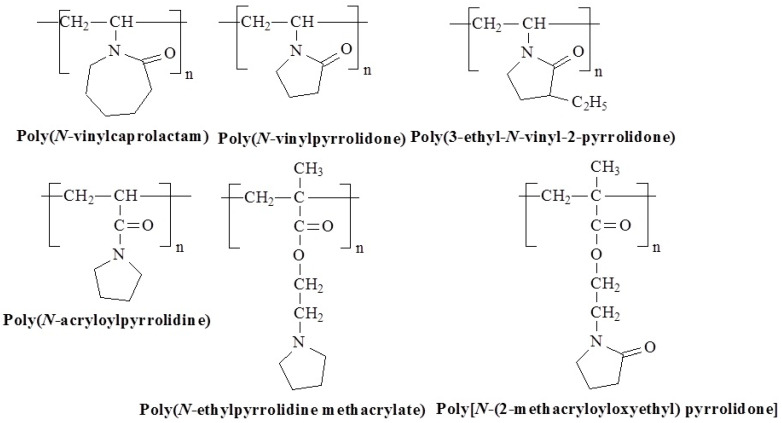
Chemical structures of temperature-responsive lactam/pyrrolidone/pyrrolidine based polymers.

**Figure 12 polymers-17-01580-f012:**
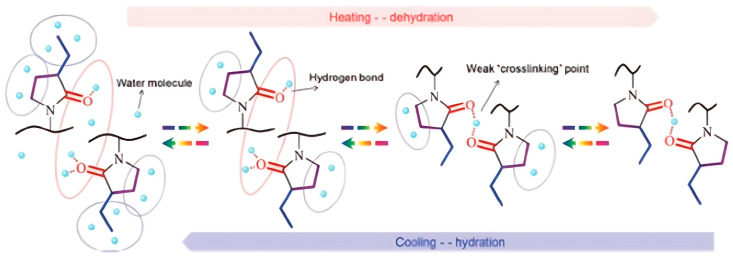
Thermoresponsive behavior of an LCST-type polymer based on a pyrrolidone structure in aqueous solution [71].

**Figure 13 polymers-17-01580-f013:**
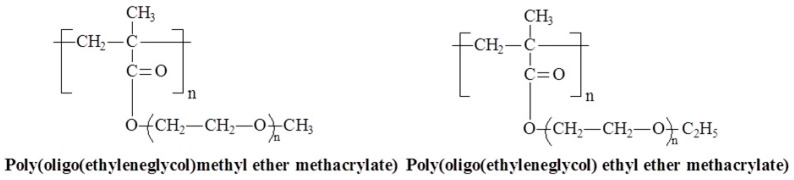
Chemical structures of temperature-responsive POEGMAs.

**Figure 14 polymers-17-01580-f014:**
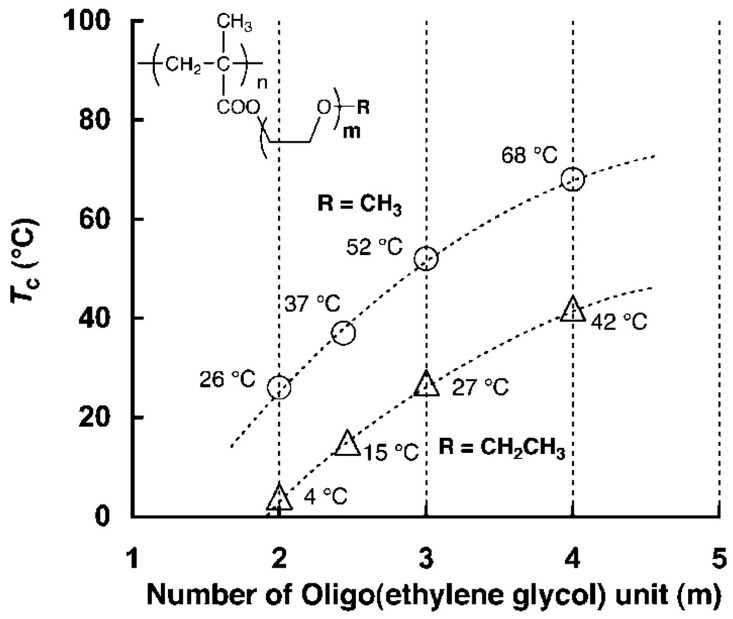
Relationship between the number of egus and LCST [79].

**Figure 15 polymers-17-01580-f015:**
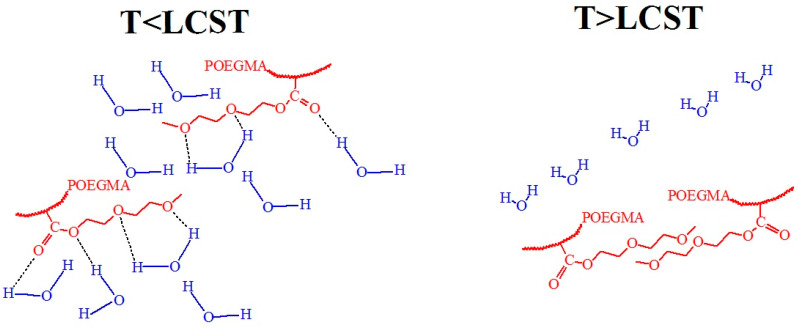
Hypothetical conformations of the hydrogen bonding between POEGMA ether oxygens and water molecules at T < LCST and van der Waals interactions at T > LCST for POEGMA.

**Figure 16 polymers-17-01580-f016:**
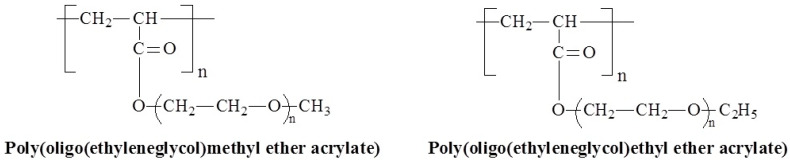
Chemical structures of temperature-responsive POEGAs.

**Figure 17 polymers-17-01580-f017:**
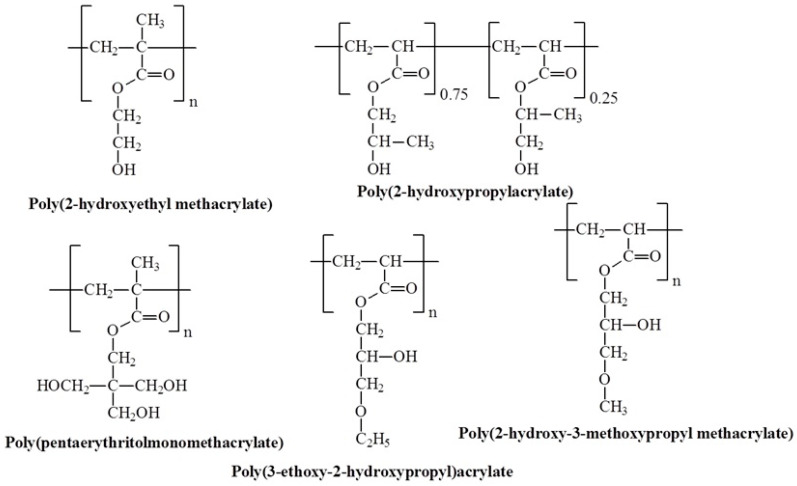
Chemical structures of temperature-responsive hydroxyl-containing polymers.

**Figure 18 polymers-17-01580-f018:**
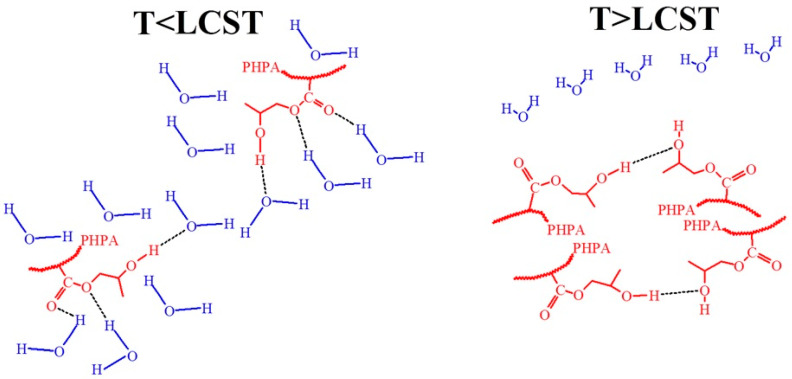
Hypothetical conformations of hydrogen bonding between hydroxyl groups and water molecules at T < LCST and hydroxyl groups in the polymer and van der Waals interactions at T > LCST for the hydroxyl-containing polymer.

**Figure 19 polymers-17-01580-f019:**
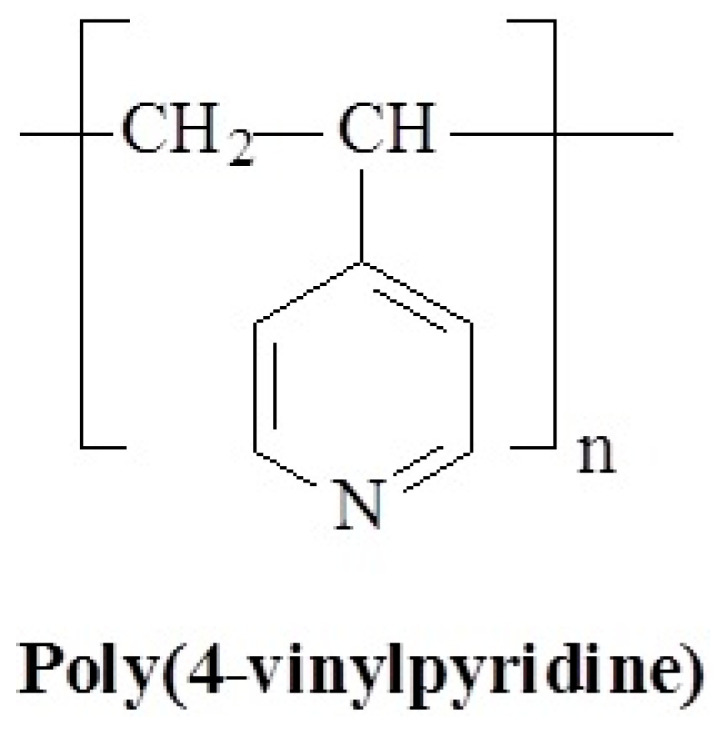
Chemical structure of poly(4-vinylpyridine).

**Figure 20 polymers-17-01580-f020:**
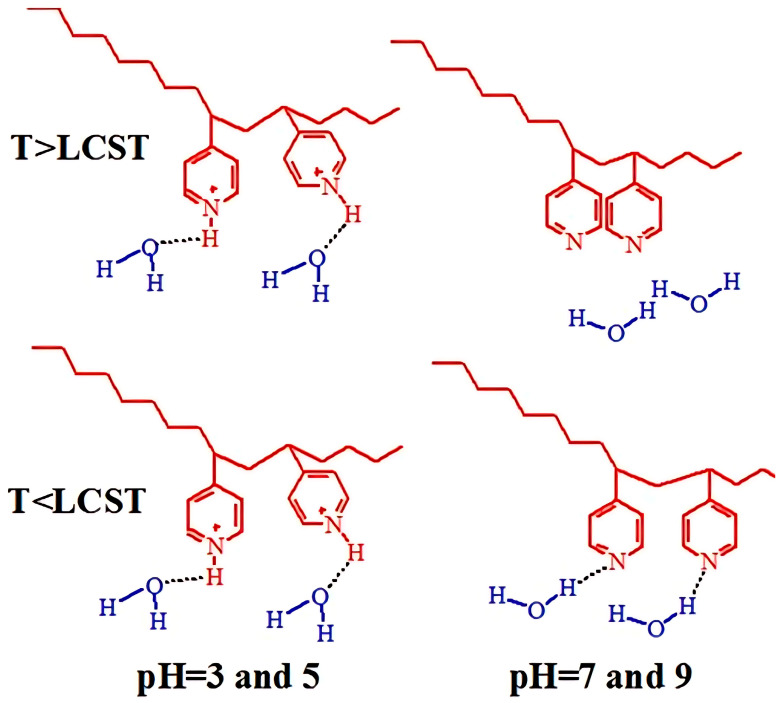
Hypothetical conformations of van der Waals interactions and hydrogen bonding among the pyridyl groups of the PVP, the protonated pyridyl groups of the P4VP, and the water molecules at various pH values and temperatures.

**Figure 21 polymers-17-01580-f021:**
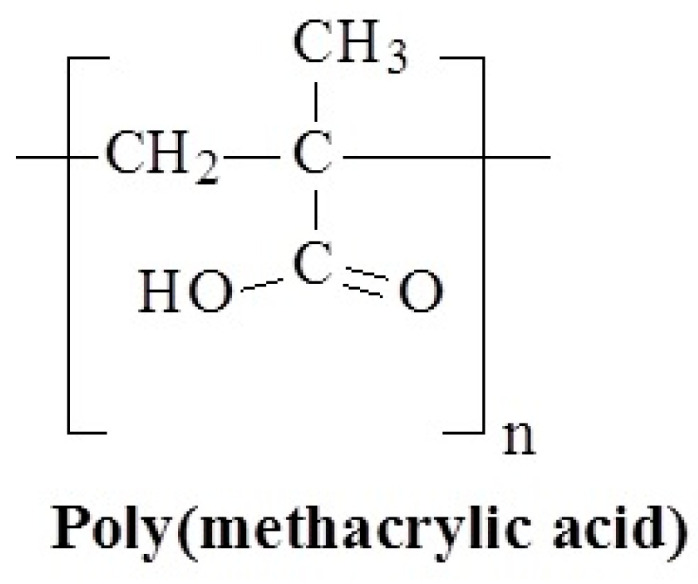
Chemical structure of poly(methacrylic acid).

**Figure 22 polymers-17-01580-f022:**
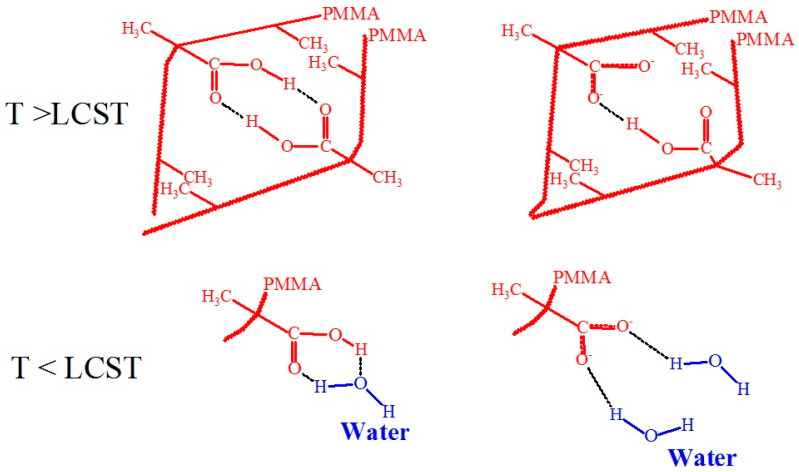
Hypothetical scheme including interactions of carboxyl or carboxylate groups with water molecules at T < LCST, polymer-polymer interactions through doubly H-bonded dicarboxylic dimers and van der Waals or hydrophobic interactions between methyl groups at T > LCST [116,118,119,120].

**Table 1 polymers-17-01580-t001:** Effects of substitution groups in poly(*n*-alkyl acrylamide)s on the LCST or solubility.

Polymer Name	Substituted Groups		LCST [°C]
	R1	R2	
Poly(*N*-alkylacrylamide)s			
poly(*N*-methylacrylamide)	H	Me	soluble
poly(*N*-ethylacrylamide)	H	Et	50
poly(*N*-cyclopropylacrylamide)	H	cPr	45
poly(*N*-isopropylacrylamide)	H	iPr	30–36
poly(*N*-n-propylacrylamide)	H	nPr	21–25
poly(*N*-n-butylacrylamide)	H	nBu	insoluble
poly(*N*-isobutylacrylamide)	H	iBu	insoluble
poly(*N*-sec-butylacrylamide)	H	sBu	insoluble
poly(*N*-tert-butylacrylamide)	H	tBu	insoluble
Poly(*N,N*-dialkylacrylamide)s			
poly(*N,N*-dimethylacrylamide)	Me	Me	soluble
poly(*N*-methyl-*N*-ethylacrylamide)	Me	Et	56–70
poly(*N*-methyl-*N*-isopropylacrylamide)	Me	iPr	22
poly(*N*-methyl-N-n-propylacrylamide)	Me	nPr	20
poly(*N,N*-diethylacrylamide)	Et	Et	32
poly(*N*-ethyl-*N*-isopropylacrylamide)	Et	iPr	insoluble
poly(*N*-ethyl-*N*-n-propylacrylamide)	Et	nPr	insoluble
poly(*N,N*-diisopropylacrylamide)	iPr	iPr	insoluble
poly(*N,N*-dipropylacrylamide)	nPr	nPr	insoluble

**Table 2 polymers-17-01580-t002:** Effects of substitution groups in poly(*n*-alkyl methacrylamide)s on the LCST or solubility [39].

Polymer Name	Substituted Groups		LCST [°C]
	R1	R2	
Poly(*N*-methylmethacrylamide)	H	Me	soluble
Poly(*N*-ethylmethacrylamide)	H	Et	soluble
Poly(*N*-cyclopropylmethacrylamide)	H	cPr	59
Poly(*N*-isopropylmethacrylamide)	H	iPr	38, 43–45
poly(*N*-n-propylmethacrylamide)	H	nPr	28
poly(*N*-n-butylmethacrylamide)	H	nBu	insoluble
poly(*N*-isobutylmethacrylamide)	H	iBu	insoluble
poly(*N*-sec-butylmethacrylamide)	H	sBu	insoluble
poly(*N*-tert-butylmethacrylamide)	H	tBu	insoluble

**Table 3 polymers-17-01580-t003:** Effects of substitution groups and ethylene glycol units (egu)s in POEGMA on the LCST or solubility [77,79].

Polymer Name	R	Number Egus	Abbreviation	LCST [°C]
Poly(di(ethylene glycol)methyl ether methacrylate)	Me	2	POEG_2_MEMA	22, 26
Poly(oligo(ethyleneglycol)_3_methyl ether methacrylate	Me	3	POEG_3_MEMA	52
Poly(oligo(ethyleneglycol)_4_methyl ether methacrylate	Me	4	POEG_4_MEMA	61, 68
Poly(oligo(ethyleneglycol)_8.5_methyl ether methacrylate	Me	8–9	POEG_8.5_MEMA	90
Poly(oligo(ethyleneglycol)_22_methyl ether methacrylate	Me	22	POEG_22_MEMA	soluble
Poly(di(ethylene glycol)ethyl ether methacrylate)	Et	2	POEG_2_EEMA	4
Poly(oligo(ethyleneglycol)_3_ethyl ether methacrylate)	Et	3	POEG_3_EEMA	26
Poly(oligo(ethyleneglycol)_4_ethyl ether methacrylate	Et	4	POEG_4_EEMA	42

**Table 4 polymers-17-01580-t004:** Effects of substitution groups and ethylene glycol units (egu)s in POEGAs on the LCST [85].

Polymer Name	R	Number Egus	Abbreviation	LCST [°C]
Poly(ethylene glycol)methyl ether acrylate)	Me	1	POEG_1_MEA	<0, 5
Poly(di(ethylene glycol)methyl ether acrylate)	Me	2	POEG_2_MEA	40
Poly(oligo(ethyleneglycol)_3_methyl ether acrylate	Me	3	POEG_3_MEA	70
Poly(oligo(ethyleneglycol)_8.5_methyl ether acrylate	Me	8–9	POEG_8.5_MEA	92
Poly(di(ethylene glycol)ethyl ether acrylate)	Et	2	POEG_2_EEA	10, 16.5
Poly(oligo(ethyleneglycol)_3_ethyl ether acrylate)	Et	3	POEG_3_EEA	34, 39

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
