# Peer review of "Molecular Design and Role of the Dynamic Hydrogen Bonds and Hydrophobic Interactions in Temperature-Switchable Polymers: From Understanding to Applications"

_polymers, 2025, doi:10.3390/polym17111580_

Round 1
Reviewer 1 Report
Comments and Suggestions for Authors
The manuscript reviews the molecular mechanisms driving the UCST/LCST based thermoresponsive transition behaviors. The hydrogen bonding and van der Waals interactions are thoroughly described to well understand the transition behaviors. The obtained mechanism can be used for the function design of thermoresponsive polymers in the near future.
However, the hydrogen bonding in the POEGMA based thermoresponsive polymers presented in Figure 15 may not correct. The influence of additional methyl group to the LCST in POEGMA and POEGA presented in Table 3 and 4 is confusing. The reviewer also wonders about the hydrogen bonding presented in Figure 9. Based on these drawbacks, the reviewer suggests a major revision before publishing this manuscript in Polymers.
1. The hydrogen bonding in the POEGMA based thermoresponsive polymers in Figure 15 may not correct. According to the previous publication, the hydrogen bonding is preferred to form between C=O and water molecules (Zhong, Q.; Metwalli, E.; Rawolle, M.; et al. Macromolecules 2013, 46, 4069-4080). The author should clarify this point in the revised manuscript.
2. The influence of additional methyl group to the LCST in POEGMA and POEGA presented in Table 3 and 4 is confusing. According to the previous publication, the additional methyl groups will induce the increase of LCST in the acrylamide based thermorepsonsive polymers (Nieuwenhuis, S.; Zhong, Q.; Metwalli, E.; et al. Langmuir 2019, 35, 24, 7691-7702.) However, a lower LCST is observed when comparing the LCST with or without methyl group with the same side chain. The author should address this point in the revised manuscript.
3. The reviewer also wonders about the hydrogen bonding presented in Figure 9. Comparing to connecting both C=O and N-H groups by one water molecule, is it more feasible that two water molecules connected to C=O and N-H groups separately?
Author Response
Responses to the Reviewer #1 comments:
The manuscript reviews the molecular mechanisms driving the UCST/LCST based thermoresponsive transition behaviors. The hydrogen bonding and van der Waals interactions are thoroughly described to well understand the transition behaviors. The obtained mechanism can be used for the function design of thermoresponsive polymers in the near future.
However, the hydrogen bonding in the POEGMA based thermoresponsive polymers presented in Figure 15 may not correct. The influence of additional methyl group to the LCST in POEGMA and POEGA presented in Table 3 and 4 is confusing. The reviewer also wonders about the hydrogen bonding presented in Figure 9. Based on these drawbacks, the reviewer suggests a major revision before publishing this manuscript in Polymers.
Remark 1. The hydrogen bonding in the POEGMA based thermoresponsive polymers in Figure 15 may not correct. According to the previous publication, the hydrogen bonding is preferred to form between C=O and water molecules (Zhong, Q.; Metwalli, E.; Rawolle, M.; et al. Macromolecules 2013, 46, 4069-4080). The author should clarify this point in the revised manuscript.
Reply 1. Thank you very much for this valuable comment. We have added the appropriate information to the main text. Additionally, we would like to note that some authors have proposed alternative mechanisms, and this issue may need further clarification in future studies. Accordingly, we have revised the relevant text and slightly modified Figure 15.
Now: “Interesting results were presented in [83, 84], where it was shown that for POEGMAs, hydrogen bonds form exclusively between the carbonyl oxygens and the hydrogen atoms of water at temperatures below the LCST. Above the LCST, most of these hydrogen bonds are disrupted; however, a certain amount of bound water remains within the polymer. This suggests that the hydrophilic layer of water surrounding the polymer fragments likely plays a key role in retaining water. Various proposed schemes describing different hydrogen bonding conformations highlight the need for further studies to clarify the exact nature and dynamics of these interactions.”
Remark 2. The influence of additional methyl group to the LCST in POEGMA and POEGA presented in Table 3 and 4 is confusing. According to the previous publication, the additional methyl groups will induce the increase of LCST in the acrylamide based thermorepsonsive polymers (Nieuwenhuis, S.; Zhong, Q.; Metwalli, E.; et al. Langmuir 2019, 35, 24, 7691-7702.) However, a lower LCST is observed when comparing the LCST with or without methyl group with the same side chain. The author should address this point in the revised manuscript.
Reply 2. The paper written by Nieuwenhuis, S.; Zhong, Q.; Metwalli, E.; et al. Langmuir 2019, 35(24), 7691–7702 includes a comparison of PNIPAM with PNIPMAM, which contains one additional methyl group in the main chain. The information we presented is fully consistent with the results reported in that paper. We have incorporated this information into the main text.
“In similar work [62], it was shown that compared to PNIPAM, PNIPMAM exhibits similar hydration capacity but a higher phase transition temperature, increasing from 29 °C to 38 °C. This suggests that while the extra methyl group does not significantly affect the overall hydration, it restricts conformational flexibility during the thermal transition.”
Also in this paper was compared properties of the PNIPMAM and poly(methoxy diethylene glycol acrylate) (in case of our nomenclature - POEG2MEA). We added appropriated text in the paper.
“In the work [62] were compared hydration and dehydration kinetics of PNIPMAM and POEG2MEA films. PNIPMAM and POEG2MEA films differ significantly in their hydration and dehydration behavior under thermal stimuli. PNIPMAM shows higher hydration capacity due to N−H and C=O groups but hydrates more slowly because of its higher glass transition temperature. Upon heating, both films undergo shrinkage, rearrangement, and partial reswelling, with PNIPMAM responding more quickly and strongly. POEG2MEA lacks N−H groups, leading to lower hydration and different interfacial behavior.”
Unfortunately we didn’t find information on a lower LCST observed when comparing the LCST with or without methyl group with the same side chain in this and similar publication. In contrast, previously published papers showed contrary results (see table below). If reviewer will provide this information in details we will include it to manuscript.
|
Polymer name |
Number egus |
Abbreviation |
LCST [°C] |
|
Poly(di(ethylene glycol)methyl ether methacrylate) |
2 |
POEG2MEMA |
22, 26 |
|
Poly(di(ethylene glycol)methyl ether acrylate) |
2 |
POEG2MEA |
40 |
|
Poly(oligo(ethyleneglycol)3methyl ether methacrylate |
3 |
POEG3MEMA |
52 |
|
Poly(oligo(ethyleneglycol)3methyl ether acrylate |
3 |
POEG3MEA |
70 |
Additionally, we believe that in poly(N-alkyl methacrylamide)s, the presence of a methyl group on the methacrylamide backbone increases the LCST compared to their acrylamide counterparts. This can be attributed to the methyl group’s steric hindrance, which disrupts intra- and intermolecular hydrogen bonding among the amide groups. This disruption enhances the polymer’s interaction with water, thereby increasing hydrophilicity and raising the LCST. For example, poly(N-isopropylmethacrylamide) (PNIPMAM) exhibits a higher LCST than poly(N-isopropylacrylamide) (PNIPAM) due to this effect. Conversely, in poly(oligo(ethylene glycol) methacrylate)s (POEGMAs), the methyl group on the methacrylate backbone decreases the LCST compared to poly(oligo(ethylene glycol) acrylate)s (POEGAs). The methyl group increases the hydrophobicity of the polymer backbone, reducing the polymer’s affinity for water and thus lowering the LCST. This effect is more pronounced in polymers with shorter oligo(ethylene glycol) side chains, where the hydrophobic influence of the methyl group is not sufficiently offset by the hydrophilic side chains. In summary, the impact of the methyl group on LCST is context-dependent: it increases LCST in poly(N-alkyl methacrylamide)s by disrupting hydrogen bonding and enhancing hydrophilicity, while it decreases LCST in poly(oligo(ethylene glycol) methacrylate)s by increasing hydrophobicity.
Remark 3. The reviewer also wonders about the hydrogen bonding presented in Figure 9. Comparing to connecting both C=O and N-H groups by one water molecule, is it more feasible that two water molecules connected to C=O and N-H groups separately?
Reply 3. We completely agree with the reviewer that, in many cases, it is indeed more favorable for separate water molecules to interact individually with the C=O and N-H groups rather than for a single water molecule to bridge both. This arrangement allows for more optimal hydrogen bonding interactions, as each water molecule can orient itself to maximize bond strength and geometry with its specific functional group, thereby reducing steric hindrance and strain compared to forcing one water molecule into an energetically less favorable bridging position. Accordingly, we have revised Figure 9.
Reviewer 2 Report
Comments and Suggestions for Authors
The work is devoted to study of bonds in a temperature-responsive polymer. This topic is important because understanding the mechanisms of interaction of bonds in a temperature-responsive polymer will significantly expand their usage.
The authors provided an extensive literature analysis on the topic and, what is more important, presented their own analysis of the literature reviewed and put forward several hypotheses about the involvement of different groups in interactions (for example, the role of a carboxyl group)
What is the main question addressed by the research?
The work is devoted to the study of the role of hydrogen bonds and
hydrophobic compounds in thermopolymers.
Do you consider the topic original or relevant to the field?
As far as the article is considered as a review, so it is incorrect to say about the complete originality data. However, because a complete literature analysis and conclusions on it are mentioned by the authors in this review the work can be considered as unique and original.
Does it address a specific gap in the field? Please also explain why this is/ is not the case. What does it add to the subject area compared with other published material?
The authors analyzed an extended volume of literature data. The materials presented in the article allow you to navigate in the issue related to usage of thermopolymers much better. Moreover, this review allows a better understanding of the mechanisms of interaction of groups in thermopolymers. I would like to make a special note that the
authors provide the conclusions but not just a massive amount of data.
This is what makes this review stand out from many other works. In my opinion, currently, there is a pernicious tendency to consider reviews as a summary of literary data, which is fundamentally wrong and a review requires the author's personal reworking of this data, indicating the weaknesses in the analyzed topic.
What specific improvements should the authors consider regarding the
methodology?
The methodology of storytelling is logical in the article.
Are the conclusions consistent with the evidence and arguments presented
and do they address the main question posed? Please also explain why this
is/is not the case.
The authors provided practical usage of thermopolymers and presented
the role of hydrogen and van der Waals bonds in the thermosensitivity
of these polymers.
Are the references appropriate?
The references cover as classical as modern literature.
Any additional comments on the tables and figures.
On the pic 4 there are the designations, maybe they are from other picture (the letters a,b,c are marked in green).
Author Response
Responses to the Reviewer #2 comments:
The work is devoted to study of bonds in a temperature-responsive polymer. This topic is important because understanding the mechanisms of interaction of bonds in a temperature-responsive polymer will significantly expand their usage.
The authors provided an extensive literature analysis on the topic and, what is more important, presented their own analysis of the literature reviewed and put forward several hypotheses about the involvement of different groups in interactions (for example, the role of a carboxyl group)
Remark 1. On the pic 4 there are the designations, maybe they are from other picture (the letters a,b,c are marked in green).
Reply 1. Figure 4 was corrected
Reviewer 3 Report
Comments and Suggestions for Authors
The study entitled “Molecular design and role of the dynamic hydrogen bonds and hydrophobic interactions in temperature-switchable polymers: from understanding to applications” by Stetsyshyn et al. consolidates the studies concerning the molecular design and role of the dynamic hydrogen bonds and hydrophobic interactions in the temperature switchable polymers. The consolidation has been very well carried out and the authors have carried out the extensive literature survey in the field. The references have been adequately discussed. The figures have been well presented except few (Like figures 12 and 14 which seem to be a little blurry. I would suggest some clarification in these figures in addition to a spelling mistake “Preferebly” in figure 4. This review article is going to be a value addition in the temperature-switchable polymers literature. I highly recommend the publication of this manuscript.
The study entitled “Molecular design and role of the dynamic hydrogen bonds and hydrophobic interactions in temperature-switchable polymers: from understanding to applications” by Stetsyshyn et al. consolidates the studies concerning the molecular design and role of the dynamic hydrogen bonds and hydrophobic interactions in the temperature switchable polymers. The consolidation has been very well carried out and the authors have carried out the extensive literature survey in the field. The references have been adequately discussed. The figures have been well presented except few (Like figures 12 and 14 which seem to be a little blurry. I would suggest some clarification in these figures in addition to a spelling mistake “Preferebly” in figure 4. This review article is going to be a value addition in the temperature-switchable polymers literature. I highly recommend the publication of this manuscript.
- What is the main question addressed by the research?
Since it is a review article, the dearth of literature consolidation and discussion in the field of temperature-switchable polymers from the molecular design perspective seems to have been filled by this article.
• Do you consider the topic original or relevant to the field? Does it
address a specific gap in the field? Please also explain why this is/ is not
the case.
The article address the literature gap in the relevant field appropriately.
• What does it add to the subject area compared with other published
material?
It summarizes the reports of temperature-sensitive polymers from a molecular design perspective through hydrogen bond and hydrophobic interactions which is something reports of which are scarce in the literature.
• What specific improvements should the authors consider regarding the
methodology?
Since it is a review article, so experimental methodology question doesn’t arise.
• Are the conclusions consistent with the evidence and arguments presented
and do they address the main question posed? Please also explain why this
is/is not the case.
Yes, the conclusions are well presented and they have listed out the future directions of the field as well with some potential questions.
• Are the references appropriate?
Yes.
• Any additional comments on the tables and figures.
No.

Author Response
Responses to the Reviewer #3 comments:
The study entitled “Molecular design and role of the dynamic hydrogen bonds and hydrophobic interactions in temperature-switchable polymers: from understanding to applications” by Stetsyshyn et al. consolidates the studies concerning the molecular design and role of the dynamic hydrogen bonds and hydrophobic interactions in the temperature switchable polymers. The consolidation has been very well carried out and the authors have carried out the extensive literature survey in the field. The references have been adequately discussed.
Remark 1. The figures have been well presented except few (Like figures 12 and 14 which seem to be a little blurry. I would suggest some clarification in these figures in addition to a spelling mistake “Preferebly” in figure 4.
Reply 1. Figure 4 was corrected. Quality of the figures 12 and 14 was improved.
Round 2
Reviewer 1 Report
Comments and Suggestions for Authors
The reviewer appreciates the reply from the authors. All the questions and comments are well answered. Therefore, the reviewer believes that the manuscript is ready to publish in Polymers.